# High precision hyperfine measurements in Bismuth challenge bound-state strong-field QED

Johannes Ullmann[1,2,3], Zoran Andelkovic[4], Carsten Brandau[4,5], Andreas Dax[6], Wolfgang Geithner[4], Christopher Geppert[1,7], Christian Gorges[1], Michael Hammen[7,8], Volker Hannen[9], Simon Kaufmann[1], Kristian König[1], Yuri A. Litvinov[4], Matthias Lochmann[1], Bernhard Maaß[1], Johann Meisner[10], Tobias Murböck[11], Rodolfo Sánchez[4], Matthias Schmidt[10], Stefan Schmidt[1], Markus Steck[4], Thomas Stöhlker[2,3,4], Richard C. Thompson[12], Christian Trageser[5], Jonas Vollbrecht[9], Christian Weinheimer[9] & Wilfried Nörtershäuser[1]

Electrons bound in highly charged heavy ions such as hydrogen-like bismuth $^{209}Bi^{82+}$ experience electromagnetic fields that are a million times stronger than in light atoms. Measuring the wavelength of light emitted and absorbed by these ions is therefore a sensitive testing ground for quantum electrodynamical (QED) effects and especially the electron–nucleus interaction under such extreme conditions. However, insufficient knowledge of the nuclear structure has prevented a rigorous test of strong-field QED. Here we present a measurement of the so-called specific difference between the hyperfine splittings in hydrogen-like and lithium-like bismuth $^{209}Bi^{82+,80+}$ with a precision that is improved by more than an order of magnitude. Even though this quantity is believed to be largely insensitive to nuclear structure and therefore the most decisive test of QED in the strong magnetic field regime, we find a 7-$\sigma$ discrepancy compared with the theoretical prediction.

[1] Institut für Kernphysik, Technische Universität Darmstadt, Schlossgartenstraße 9, 64289 Darmstadt, Germany. [2] Helmholtz Institut Jena, Fröbelstieg 3, 07743 Jena, Germany. [3] Institut für Optik und Quantenelektronik, Friedrich-Schiller-Universität, Max-Wien-Platz 1, 07743 Jena, Germany. [4] GSI Helmholtzzentrum für Schwerionenforschung, Planckstraße 1, 64291 Darmstadt, Germany. [5] I.Physikalisches Institut, Justus-Liebig-Universität Gießen, Heinrich-Buff-Ring 16, 35392 Gießen, Germany. [6] Paul Scherrer Institut, 5232 Villigen, Switzerland. [7] Institut für Kernchemie, Johannes Gutenberg-Universität Mainz, Fritz-Strassmann-Weg 2, 55128 Mainz, Germany. [8] Helmholtz Institut Mainz, Johannes Gutenberg-Universität Mainz, Staudingerweg 18, 55128 Mainz, Germany. [9] Institut für Kernphysik, Westfälische Wilhelms-Universität Münster, Wilhelm-Klemm-Straße 9, 48149 Münster, Germany. [10] Physikalisch-Technische Bundesanstalt, Bundesallee 100, 38116 Braunschweig, Germany. [11] Institut für Angewandte Physik, Technische Universität Darmstadt, Schlossgartenstraße 7, 64289 Darmstadt, Germany. [12] QOLS Group, Department of Physics, Imperial College London, Prince Consort Road, South Kensington, London SW7 2AZ, UK. Correspondence and requests for materials should be addressed to W.N. (email: wnoertershaeuser@ikp.tu-darmstadt.de).

Even a single electron in vacuum cannot be thought of as an isolated particle, because it interacts with itself and with the vacuum around it, being filled with virtual particle–antiparticle pairs. The theory of quantum electrodynamics (QED) has been developed to take all these effects into account and was confirmed to very high precision for free electrons[1] and electrons bound to light nuclei[2]. In hydrogen, the spin of the single electron in the ground state is oriented either parallel or anti-parallel to the nuclear spin. This leads to a splitting of the electronic ground state into two levels, the hyperfine structure splitting (HFS). An energy difference between the two orientations arises from the fact that the electron carries a magnetic moment that can be aligned energetically favourably (parallel) or disfavourably (antiparallel) in the magnetic field of the nucleus. The corresponding splitting energy in hydrogen is a few $\mu$eV and the transition between the two states is connected with the emission or absorption of electromagnetic waves with a wavelength of 21 cm. This line and its Doppler shift are extensively used in radio astronomy, to determine movements of stars and galaxies. The exact size of the splitting is influenced by QED effects and nuclear structure contributions. Precision measurements of the transition frequency in hydrogen have provided early tests of QED[3], which were ultimately limited by the insufficient knowledge of the proton's internal structure. All H-like ions of nuclei with non-zero nuclear spin show a similar hyperfine structure. The size of QED effects, however, rises dramatically in heavy highly charged ions, just as the experimental difficulties measuring them. The magnetic fields created by the heavy nucleus are the strongest fields available for experiments and make them a sensitive testing ground for QED effects. Owing to the high nuclear charge and the close proximity of the electron to the nucleus, the electric and magnetic fields averaged across the electron's orbital can be a million times stronger than for the electron in hydrogen. New effects might appear in the interaction of the electron with itself, the vacuum or the nuclear fields in this regime, that is, the hyperfine interaction might be affected by the existence of new particles not included yet in the current standard model and therefore not considered in state-of-the-art QED calculations.

Although measurements of X-ray transition energies in H-like[4] and Li-like[5] $^{238}$U are mainly sensitive to the strong electric fields, measurements of the HFS are complementary, as they provide sensitivity to effects arising in the strong magnetic fields. As an example, the magnetic field at the surface of the nucleus of $^{209}$Bi exceeds $10^9$ T and the average field an 1s electron experiences is on the order of 20,000 T, a thousand times stronger than that of the strongest superconducting magnets that can be built. As the splitting energy scales with the third power of the nuclear charge, one of the largest splittings in a stable or primordial isotopes is observed in H-like $^{209}$Bi$^{82+}$ and has been measured by laser spectroscopy more than 20 years ago[6].

To test QED-based theoretical calculations in these heavy systems, the hyperfine splittings in H-like Ho, Re and Tl have been measured in electron beam ion traps[7–9], whereas Pb and Bi were studied at the experimental storage ring (ESR) at the GSI Helmholtzzentrum für Schwerionenforschung in Darmstadt[6,10]. Even though these measurements were sufficiently accurate, they could not be exploited as meaningful QED tests, due to a large uncertainty in the calculation of the distribution of magnetization in the nucleus, called the Bohr–Weisskopf (BW) effect[11]. To remove the nuclear structure uncertainty it was therefore proposed to measure the HFS in H-like ($\Delta E^{(1s)}$) and Li-like ($\Delta E^{(2s)}$) ions of the same isotope[12]. The specific difference $\Delta'E$ between these splittings

$$\Delta'E = \Delta E^{(2s)} - \xi \Delta E^{(1s)} \qquad (1)$$

is expected to provide the most decisive test of QED in the strong magnetic field regime. The parameter $\xi = 0.16886$ is chosen from theory to cancel the BW correction for $^{209}$Bi (ref. 12). The remaining uncertainties among other contributions to $\Delta'E$ are listed in Table 1. The first laser spectroscopic observation of the hyperfine structure in a Li-like heavy system was recently reported[13] but did not provide sufficient accuracy to become sensitive to QED contributions in $\Delta'E$.

Here we measured both splittings with sufficient accuracy to reliably test QED contributions to this specific difference. Compared with the theoretical predictions for this case, we find a significant discrepancy of about seven times the combined uncertainties of experiment and theory, which is the largest deviation reported in strong-field QED up to now. Our result joins a series of recent measurements that challenge the theory of QED and electron–nuclear interactions such as the muon g-factor[14], the proton radius[15] and, most recently, the charge radius of the deuteron[16].

## Results

**Experiment.** At GSI, beams of $^{209}$Bi$^{82+}$ and $^{209}$Bi$^{80+}$ were stored in the ESR (sketched in Fig. 1) at a velocity of $\approx 71\%$ of the speed of light. Owing to this relativistic velocity, the wavelengths of the HFS transitions in the ions' frame of reference $\lambda_0$ are shifted towards

$$\lambda_{\text{Lab}} = \frac{\lambda_0}{\gamma(1 + \beta \cos\theta)} \qquad (2)$$

in the laboratory frame due to the Doppler effect. Here, $\theta$ is the angle between the directions of the laser and the ion beam during the interaction, $\gamma = (\sqrt{1 - \beta^2})^{-1}$ the time dilation factor and $\beta = v/c$ the velocity $v$ in terms of the speed of light $c$. Thus, in an ion beam with substantial velocity spread, light is correspondingly absorbed at different wavelengths, leading to a broadening of the laser resonance. To reduce this effect, the ion beam is cooled by the interaction with an electron beam[17] along a path of about 2.5 m inside the electron cooler (described in Methods and shown in Fig. 1). The large Doppler shift is a mixed blessing: it makes it possible to use the same powerful state-of-the-art pulsed laser systems in the visible range to address transitions that are either in the ultraviolet or in the infrared region by changing $\theta$. However, the required large Doppler correction is a potential source for systematic uncertainties as will be discussed in the Methods section.

---

**Table 1 | Calculated contributions to the specific difference.**

| | |
|---|---|
| Dirac | $-31.809$ |
| Interelectronic Interaction $1/Z$ | $-29.995$ |
| Interelectronic Interaction $1/Z^2$ | $0.258$ |
| Interelectronic Interaction $1/Z^3$ | $-0.003(3)$ |
| Single-electron QED | $0.036$ |
| Screened QED | $0.193(2)$ |
| | |
| *Remaining uncertainties of nuclear effects:* | |
| Bohr–Weisskopf | $0.003$ |
| Nuclear magnetic moment | $0.003$ |
| Nuclear polarization | $0.002$ |
| $\Delta'E$ | $-61.320(4)(5)$ |
| $\Delta'E_{\text{experiment}}$ (this work) | $-61.012(5)(21)$ |

Theoretical results and uncertainties of $\Delta'E = \Delta E^{(2s)} - \xi \Delta E^{(1s)}$ and its individual contributions taken from ref. 25. All values are in meV. The first and second uncertainties in the total value of $\Delta'E$ arise from uncalculated higher-order terms and the uncertainty of the complete cancellation of all nuclear effects, respectively. Uncertainties of $\Delta'E_{\text{experiment}}$ are the statistical and systematic uncertainty contributions as listed in Table 2 and discussed in the Methods section.

---

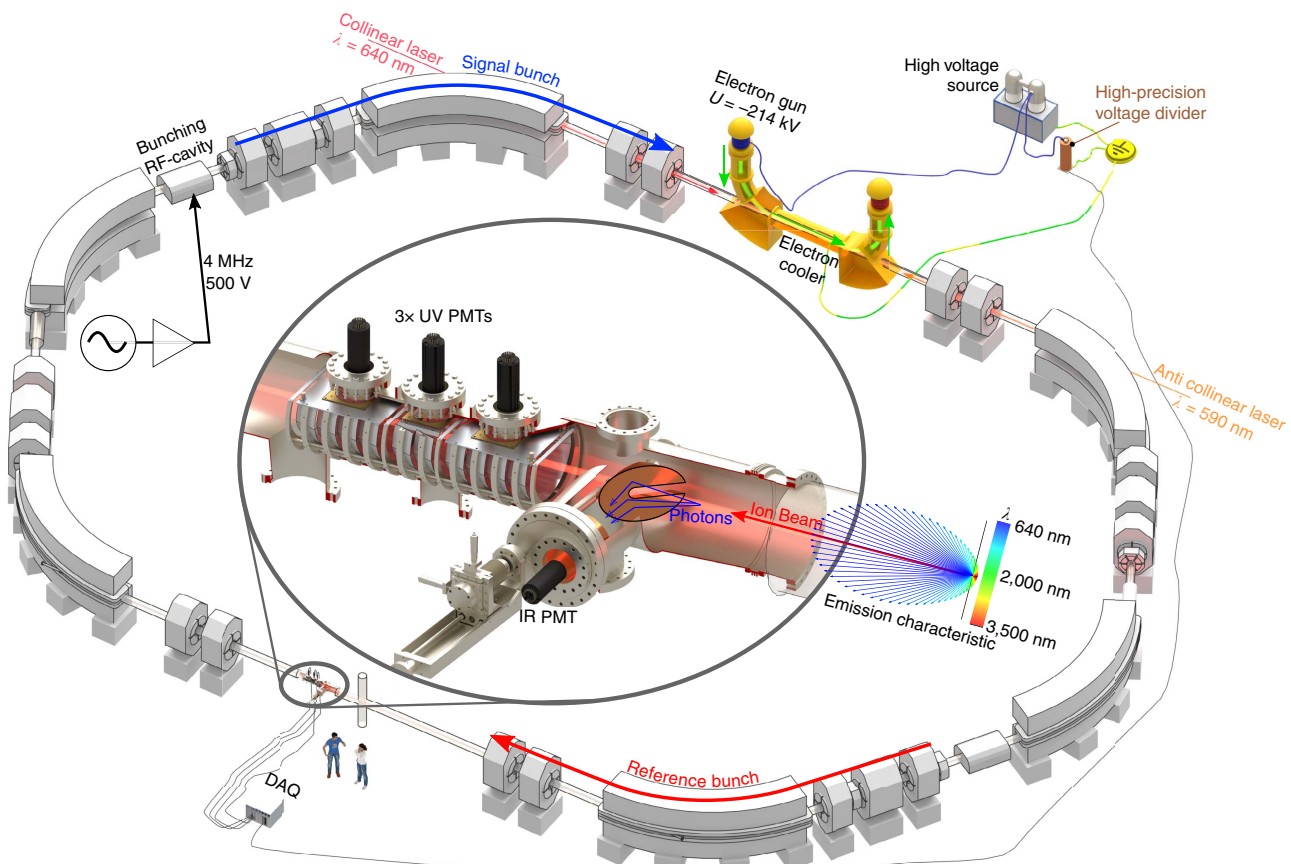

**Figure 1 | Overview of the experimental setup at GSI.** The bismuth ions revolve in the ESR and the velocity spread of the ions is reduced by electron cooling. The final velocity is determined by the accelerating potential of the electron beam, which is measured with a high-precision voltage divider. Two ion bunches are formed by applying the second harmonic of the revolution frequency to a radio frequency (RF) cavity. Fluorescence and background photons are reflected by specialized mirror systems (inset) and detected with photo multiplier tubes (PMT) for ultraviolet and infrared light. A newly designed parabolic copper mirror system[30] allowed the detection of the resonance in the Li-like charge state[13]. The data acquisition system (DAQ) comprises fast time-to-digital converters to resolve the temporal phase of each detected photon.

**Hyperfine splittings.** The observed resonances are shown in Fig. 2. The Doppler-corrected rest frame transition energies of the H-like and the Li-like ions, as listed in Table 2 and plotted in the two upper frames of Fig. 3, agree well with previous measurements, but the total uncertainty is reduced by an order of magnitude. The large scatter of the theoretical predictions for the H-like case reflects the strong model dependence of the BW correction[18–21]. For the hyperfine splitting in the Li-like ion, one has to distinguish between the *ab-initio* theoretical predictions with a similar scatter[19,22,23] and the three more accurate predictions[12,24,25] based on the measured splitting in the H-like ion[6], combined with the calculated $\Delta'E$, which is assumed to be correct. The latter are also included in the enlarged inset in the centre frame of Fig. 3.

**The specific difference.** The actual test of strong-field QED is depicted in the lowest frame of Fig. 3, where the specific difference as calculated from our measurements of the two charge states is compared with theoretical predictions. We find a very large deviation of about 0.3 meV from theory, which is about seven times (this value is based on the smallest distance between the edges of the error bars, while the central values are 8 -σ apart) the combined uncertainties of the most recent theoretical value and the experimental result. This is almost twice as large as all screened QED contributions to $\Delta'E$ and even larger than the $1/Z^2$ interelectronic interaction contribution. Hence, we assume that

there is a more fundamental problem in the calculations, which has to be identified. Possible reasons could be that the magnetic moment of the $^{209}$Bi nucleus is considerably different from the accepted value in the literature due to incorrectly determined diamagnetic-shielding or chemical-shift corrections for the NMR data or that the cancellation of the BW effect does not work as expected.

## Discussion

On the experimental side, the reason for the observed discrepancy could be scrutinized by measuring $\Delta'E$ for a different system. Here, $^{208}$Bi with a half-life of $3.7 \times 10^5$ years would be an excellent candidate. It has a different spin ($I = 5$) and a nuclear magnetic moment that is about 10% larger than that of $^{209}$Bi. The ratio of both moments[26] was re-measured recently. Contributions from the proton outside of the $Z = 82$ shell closure but also from a neutron-hole below the $N = 126$ shell closure give rise to a magnetic moment distribution and therefore the BW effect, which is considerably different from that of $^{209}$Bi. The larger splitting facilitates the detection, which is advantageous as the number of ions available in the storage ring will be at least an order of magnitude smaller than for $^{209}$Bi. This is caused by the need to produce this isotope in a nuclear reaction, before it can be injected into the ESR. With such a measurement one could discriminate between the two explanations given above for the observed discrepancy: in both cases, we would observe again a

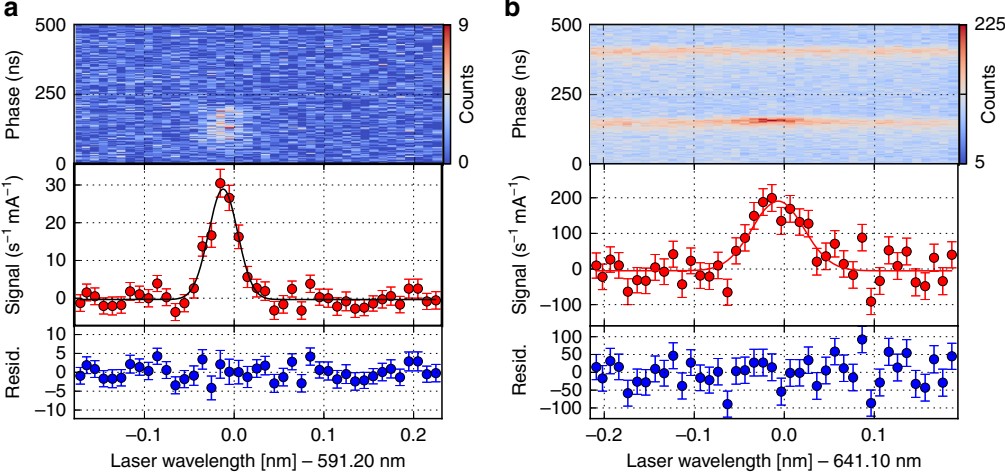

**Figure 2 | Resonances of H-like and Li-like bismuth.** Typical resonance spectra in a coasting (unbunched) H-like $Bi^{82+}$ ion beam (**a**) and a bunched Li-like $Bi^{80+}$ ion beam (**b**). In each figure, the topmost part shows the fluorescence counts in colour code as a function of the laser wavelength ($x$ axis) and the temporal phase of the photon detection ($y$ axis) with respect to the revolution frequency ($\approx 2$ MHz). The span of the ordinate equals one revolution period in the storage ring. Underneath are shown the normalized fluorescence spectrum and the residual of the weighted Gaussian fit (solid line) at the bottom. All graphs share the same abscissa. The error bars represent the statistical uncertainty. For more details, see text.

**Table 2 | Estimation of experimental uncertainties.**

| | $^{209}Bi^{82+}$ | | | $^{209}Bi^{80+}$ | | |
|---|---|---|---|---|---|---|
| | **Correction** | **Statistical** | **Systematic** | **Correction** | **Statistical** | **Systematic** |
| Gaussian fit | | 2.9 | | | 5.0 | |
| Wavelength measurement | + 3.1 | 0.4 | 1.2 | − 0.5 | 0.3 | 1.1 |
| Bunching amplitude | | | 7.4 | | | 7.3 |
| Ion current | | | 11.7 | | | 11.7 |
| Space charge | − 73.9 | | 6.8 | − 85.6 | | 6.8 |
| High-voltage measurement | | 1.4 | 7.0 | | 1.1 | 7.0 |
| Overlap angle | | | + 4.9 | | | − 0.8 |
| Total uncertainty (p.p.m.) | | 3.3 | 17.7 | | 5.2 | 17.0 |
| Final value $\lambda_0$ | | 243.8221 (8)(43) nm | | | 1554.377 (8)(28) nm | |
| Final value $\Delta E$ | | 5085.03 (2)(9) meV | | | 797.645 (4)(14) meV | |

Corrections for systematic shifts of the rest frame transition wavelength $\lambda_0$ and contributions to its statistical and systematic uncertainty caused by various experimental parameters as discussed in the Methods section. All values are provided in p.p.m. of $\lambda_0$ and represent 1-$\sigma$ confidence regions. Final values for $\lambda_0$ and the hyperfine splitting energy are given with statistical and systematic 1-$\sigma$ uncertainties in parentheses.

deviation from the predicted $\Delta'E$—but in the first case the deviation would scale exactly with the magnetic moment, whereas an incomplete compensation of the BW effect would show a different scaling. A measurement of $^{207}Bi$—with identical spin and almost identical nuclear moment as $^{209}Bi$—is an alternative candidate, which, however, does not allow for such a distinct discrimination as $^{208}Bi$, as the nuclear magnetic moment arises mainly from the unpaired proton as in $^{209}Bi$.

This discussion shows that measurements of magnetic moments not affected by diamagnetic shielding and chemical shift corrections, and their concomitant uncertainties are very important for fundamental physics investigations. Assuming that all the atomic structure calculations are correct, one can extract the required size of the nuclear moment that is in accordance with our result. However, at this point we refrain from doing so, as an independent test of the underlying theory should be carried out first. In the long run, the nuclear moment of $^{209}Bi$ will be measured on the H-like ion in the ARTEMIS[27] Penning trap at the HITRAP facility at GSI and might then provide important additional experimental input.

The hyperfine puzzle that has been established with our measurement has the potential to challenge our understanding of the physics in strong nuclear fields or the electron–nuclear interaction. Along with the overdue re-determination of the nuclear magnetic moment of $^{209}Bi$ and measurements of the specific difference in $^{208}Bi$, high precision measurements in the ion traps SpecTrap[28,29] and ARTEMIS[27] will provide an important test of our result with completely different systematics and further insights into bound-state strong-field QED.

## Methods

**Laser spectroscopy at the storage ring ESR.** At the GSI Helmholtzzentrum für Schwerionenforschung in Darmstadt, heavy, highly charged ions are produced in a stepwise process by removing all but a few electrons from an atom. Therefore, lowly charged ions of low charge states are accelerated and then sent through thin foils or gas jets at high speed. In our case, Li-like $Bi^{80+}$ and H-like $Bi^{82+}$ are produced at a velocity of $\approx 71\%$ of the speed of light and are then injected into the ESR, which is shown schematically in Fig. 1. Typically, $10^8$ ions are circulating in the ESR (circumference $\approx 108$ m) with a revolution frequency of $\approx 2$ MHz. To perform high-resolution laser spectroscopy, the velocity distribution of the ions is reduced by electron cooling[17] after injection and their average velocity is precisely determined to account for the Doppler shift in the interaction with laser light (see equation 2).

To resonantly excite the ions, laser pulses of 10 ns temporal length and up to 150 mJ pulse energy are superposed with the ion beam along the straight section at the electron cooler. For H-like Bi they are counterpropagating ($\theta = \pi$ in equation 2) to shift the absorption wavelength of the ultraviolet transition from 243 to 590 nm in the laboratory frame, whereas for Li-like Bi the copropagating geometry ($\theta = 0$) is chosen to shift the infrared transition from 1,554 to 640 nm, accordingly. To

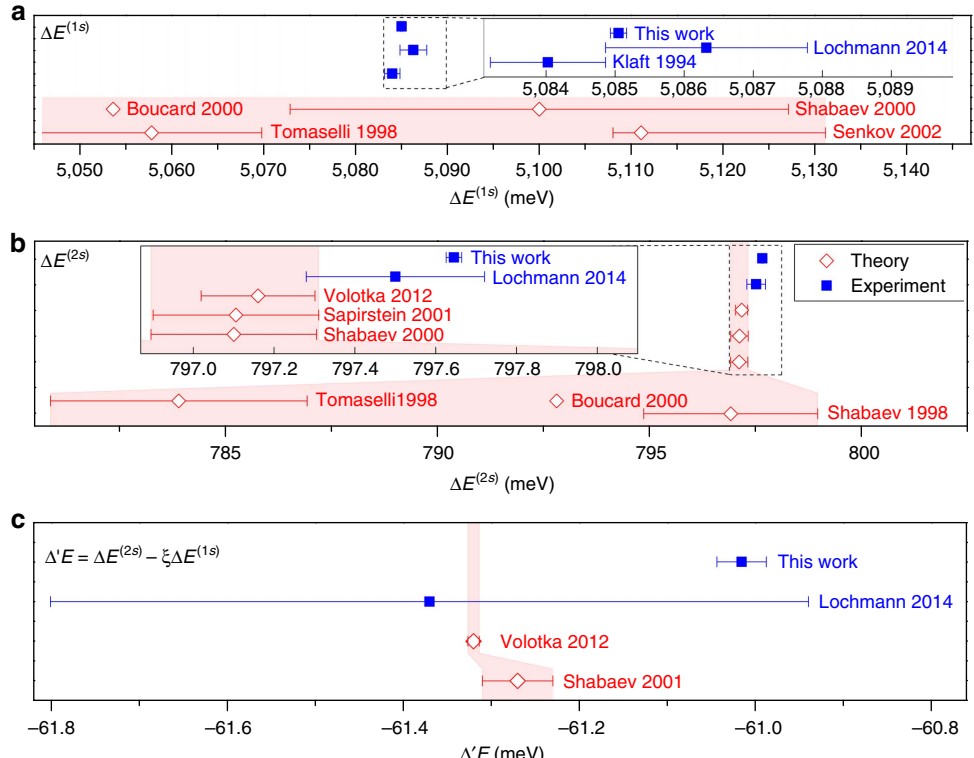

**Figure 3 | Size of hyperfine splittings and the specific difference. (a)** Evolution of the HFS energies in H-like bismuth, obtained experimentally (blue solid squares)[6,13] and from theoretical predictions (red open diamonds)[19-21,23]. **(b)** The transition in Li-like bismuth was measured by ref. 13 following various predictions[19,22,23,35] (inset:[20,24,25]). The specific difference $\Delta'E$ is shown in **c** as predicted from theory[12,25] and obtained in experiments[13]. The error bars are taken from the respective publication, systematic and statistical experimental uncertainties were added linearly.

enhance the laser-ion interaction, a radio frequency cavity at the ESR is driven at the second harmonic of the ions' revolution frequency with an amplitude of 500 V. This leads to the formation of two ion bunches of $\approx 10$ m length that revolve synchronized with the radio frequency. One of these, the signal bunch, is repetitively excited by the laser pulses, whereas the reference bunch never interacts with the laser but provides a background signal of fluorescence produced by collisions of the ions with residual gas.

Fluorescence emission after laser excitation is distributed across the complete circumference of the storage ring, due to the long lifetimes of the excited hyperfine states, which are on the order of milliseconds. The signal is detected on the opposite side of the storage ring, far away from any stray light of the laser pulse, using specialized mirror systems[30,31] (inset in Fig. 1), which reflect the fluorescence photons onto photo multiplier tubes. A real-time data acquisition system records the time of each photon event with a 3.3 ns temporal resolution. The resonance is observed by tuning the laser wavelength, measured with an interferometer, in 0.01 nm steps across the resonance position.

**Resonance signal.** The raw signal of the fluorescence is illustrated in the upper part of Fig. 2, with the number of photo multiplier events within a fixed time interval colour coded and plotted as a function of the laser wavelength on the $x$ axis and the appearance time of the photon relative to a fixed phase of the ions' revolution period on the $y$ axis. Figure 2a,b are signals for H-like and Li-like ions, respectively. The upper horizontal bands in Fig. 2b mark the background produced by collisions of the bunches with residual gas in front of the photomultipliers. Resonant interaction with subsequently enhanced fluorescence emission appears only in the signal bunch (lower band). Plotting the difference between this signal rate and the one in the reference bunch (upper band), normalized for detector dead time and ion current, results in the resonance signal plotted in the lower part. The signal shape is well reproduced with a Gaussian lineshape as indicated by the structureless fitting residuals shown in the lowest frame of each sub-figure. This was expected for an ion beam described by a Maxwell–Boltzmann velocity distribution established by intra-beam scattering and interactions with the electron beam. Figure 2a has been taken without bunching (this mode is called coasting beam), but the laser pulses were still in synchronization to the free revolution frequency. The background is now distributed equally across the complete revolution time but laser excitation is restricted to a small portion of the ions, namely those which are travelling through the straight section on the electron cooler side, while the laser pulse passes through. In this regime, the ion beam is not

confined in a bunch and the signal-to-background ratio is significantly smaller, but the beam is also not affected by the bunching frequency, which leads to a slightly narrower line than in bunching mode. The resonance frequencies obtained in bunched and coasting beam modes in the laboratory frame do agree within their statistical uncertainties. However, we have added a systematic uncertainty for the bunching effect of the size of the difference between the two results, just in case a systematic shift is covered by statistics. In case of the Li-like system the uncertainty has been scaled with the corresponding ratios of the transition wavelengths.

To obtain the transition wavelength in the rest frame of the relativistically moving ions, the Doppler correction according to equation (2) has to be applied, requiring a precise knowledge of the ion velocity $\beta$ in units of the speed of light $c$. This is determined by the electron velocity in the electron cooler[17]: here, electrons emitted from a cathode are accelerated with the electron-cooler voltage $U_{EC}$ to the velocity

$$\beta(U_{EC}) = \left[1 - \left(1 + \frac{eU_{EC}}{m_e c^2}\right)^{-2}\right]^{1/2} \quad (3)$$

and superposed with the ion beam across a length of about 2.5 m. Owing to Coulomb collisions between the electrons and the ions, a friction force, the so-called cooling force, is exerted, which impresses the average velocity of the electrons on the ions. Electron cooling needs to be maintained, to counteract energy loss of the ion beam in the residual gas of the storage ring and to counteract intra-beam scattering of the ions, that is, a self-heating of the beam. It is important to note that the electrons are continuously replaced with fresh electrons from the cathode having constant velocity and therefore this process leads to an adaption of the ion velocity to the electron velocity. A measurement of the electron cooler voltage thus provides the required information on $\beta$ for the Doppler transformation into the ions' rest frame. In our case the cooler voltage was $\approx 214$ kV and a reliable measurement of such a high voltage was not available during the previous beamtime, leading to the dominant systematic uncertainty reported in refs 13,32. Here, this issue was resolved by an *in-situ* measurement using an accurate high-voltage divider built at PTB Braunschweig, the German National Metrology Institute[33]. Having largely eliminated the dominating uncertainty from all previous measurements, a careful evaluation of all other uncertainty contributions and corrections was carried out. They are summarized in Table 2, where all contributions are presented with their effect on the final wavelength of the corresponding transition.

**Systematic uncertainties.** Besides the applied voltage, one has to consider an additional contact potential between the cathode and the drift tubes of up to 3 V, which is included in the voltage measurement systematic uncertainty. The determination of space-charge effects of the electron beam and the ion beam are conducted according to ref. 34. In the electron beam, the space charge screens electrons in the centre of the beam from the potential of the drift tubes in the cooler and therefore reduces the effective acceleration potential. The positively charged ions' space charge has an opposite effect. Both contributions have been determined experimentally: first, the electron current was systematically varied from large values (500 mA) to the minimum value that is still sufficient to cool the beam (50 mA). This leads to a change in revolution frequency observed with the Schottky diagnostics, which was then compensated by a change in the acceleration potential. Extrapolating this change to zero electron current provides the total space-charge correction. Second, a variation of the ion beam intensity did result in a very small but statistically significant change. Therefore, we have conservatively added the difference between the usual measurements at higher ion currents and the extrapolation to zero ion beam current as a systematic uncertainty. The resulting space-charge potential is in excellent agreement with a theoretical consideration of concentric cylindrical beams[34]. The Fizeau-interferometer-based laser wavelength measurement was regularly checked with a stabilized He-Ne-Laser during the measurements and finally calibrated using iodine absorption lines after completion of the measurements. A possibly undetected asymmetry in the laser lineshape was estimated as a quarter of the laser linewidth and is included in the systematic uncertainty of the wavelength measurement. The laser was carefully aligned to the ion beam using scraper plates on both ends of the electron cooler drift tubes to minimize angle deviations. Between the scrapers, the longitudinal solenoid field guides the ions along the centre axis of the straight section. As the scrapers are placed inside the toroid magnets, a residual angle of 0.3 mrad between laser and ion beam remains, which is negligible compared to other uncertainties. Ion optics behind and in front of the electron cooler, however, introduce a larger overlap angle, which was investigated by analysing the coasting-beam fluorescence data. Here, the overlap region of laser and ion beam was represented in the time structure of the signal. The comparison of spectra obtained from data of different temporal segments of the ion beam in the overlap region did not show a significant trend[34]. A conservative estimate of $\theta = 2$ mrad was obtained from the largest observed wavelength difference between two beam segments. As any angle reduces the Doppler boost for both transitions (see equation (2)), the uncertainty can be treated asymmetrically. Further uncertainties are introduced by ion beam bunching due to a possible mismatch of bunching frequency and electron cooler voltage. To evaluate this effect, the resonance wavelength with coasting ion beam is compared to measurements with varying bunching amplitudes. The systematic uncertainty is again estimated by the difference of the extrapolated resonance position at zero bunching amplitude and the value of the coasting beam measurements. Although this procedure was only possible in the H-like charge state, we assume that the effect of the bunching is similar for both charge states and therefore we take the same relative uncertainty as used for Li-like bismuth.

In total, the uncertainties amount to 0.0051 nm for H-like Bi and 0.036 nm for Li-like bismuth, representing a relative accuracy of $2.3 \times 10^{-5}$—the most accurate transition wavelength measured in a heavy highly charged ion so far.

**Data availability.** The data sets generated during the beamtime and analysed for the current study are available from the corresponding author on reasonable request.

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

## Acknowledgements

We thank F. Bosch, B. Franzke, C. Kozhuharov, J. Kluge and T. Kühl for fruitful discussions on the experimental side and we appreciate technical support during the beam time by F. Nolden, C. Dimopoulou, J. Rossbach and D. Winters, and the GSI accelerator department. We also thank S. Minami, N. Kurz and H. Brand for their support in the development of the data acquisition system and R. Jöhren for his earlier contributions to the data acquisition and the mirror system. Many enlightening discussions with Vladimir Shabaev and Andrey Volotka are appreciated. We acknowledge financial support from the German Ministry for Education and Research under grants 05P15RDFAA, 05P15PMFAA, 05P12PMFAE, 06GI947 and 05P12R6FAN, the Helmholtz-Association under contract HGF-IVF-HCJRG-108 and the Helmholtz International Centre for FAIR (HIC for FAIR) within the LOEWE programme by the federal state Hesse. M.L., C.T. and J.U. acknowledge support from HGS-HiRe.

## Author contributions

The experiment was conceived by W.N., C. Geppert and T.S. The data acquisition system was developed by M.L., M.H. and J.U. The data analysis was developed by M.L. and J.U., and carried out by J.U. The parabolic mirror detection system was developed and

manufactured by V.H., J.V. and C.W. High-voltage measurements were performed by M. Schmidt and J.M. M. Steck was responsible for the storage ring operation. The Laser system was set up and maintained by R.S. The Schottky measurement system was set up and operated by C.B. and C.T. Z.A., A.D., W.G., C. Geppert, C. Gorges, V.H., S.K., K.K., Y.A.L., M.L., B.M., T.M., W.N., R.S., S.S., M. Steck, R.C.T., J.U. and J.V. took part in the two week long data taking run. J.U. and W.N. co-wrote the paper. W.G. created the three-dimensional drawing. The manuscript was then read, improved and finally approved by all authors.

## Additional information

**Competing interests:** The authors declare no competing financial interests.

