## [Peer Review File · Nature Communications]

Reviewers' comments:

Reviewer #1 (Remarks to the Author):

This is a very important paper which demonstrates the biggest deviation from QED so far, at the level of 7 sigma (previous deviations in the muon g-factor, proton and deuterium radii are at the smaller scale but have raised an enormous interest in the scientific community). Current work is based on the brilliant idea that the nuclear uncertainty cancels out in the combination of the single-electron and 3 electron hyperfine structures.

Whatever reason for the deviation is - new physics, or unknown problems of the QED theory, or even some unknown experimental uncertainty - the paper must be published and very carefully studied by the whole physics community. This work gives a very strong motivation for further theoretical and experimental studies.

V. Flambaum.

Reviewer #2 (Remarks to the Author):

The manuscript presents measurements of the groundstate hyperfine splittings in hydrogen-like and lithium-like ^{209}Bi using a fast ion beam laser resonance technique. A certain combination of the splittings, which is theoretically predicted to be insensitive to the nuclear structure of the ^{209}Bi nucleus (which otherwise shifts the splittings through the Bohr-Weisskopf effect), is found to differ from the theoretical prediction for this combination by about 7 times the combined experimental and theoretical uncertainties. The authors conclude that the discrepancy could indicate a problem with the theory, most likely that the cancellation of the Bohr-Weisskopf effect does not occur as expected, or else that the literature values of the nuclear magnetic moment of ^{209}Bi may be in error, (presumably because of a problem with the theory that enables nuclear magnetic moments to be obtained from measurements on neutral ^{209}Bi atoms). In the abstract, they also indicate that the discrepancy may indicate a more fundamental problem with the theory of Quantum Electrodynamics. I think this particular claim is unduly speculative, albeit making the abstract more enticing to the non-expert reader. But overall, I find the manuscript scientifically sound (but see my detailed comments), interesting, well written, and suitable for publication in Nature Communications.

Detailed comments:

Abstract:

Although most of the abstract suggests that "fundamental theory of QED" is being tested" (which in any case is a rather old fashioned choice of words, since QED, as part of electroweak theory, is no longer "fundamental"), the real issue is the theory of electron-nuclear interactions - the nucleus, or even the proton, being an enormously more complicated object than the electron. So I suggest that the last sentence in the abstract be given greater emphasis. (Line 40 should also have "charge radius of the "deuteron", not of "deuterium").

Lines 57-59: Sentence structure: There should be commas after "regime" and "does not account for". But also " does not account for, or is modified.." is unclear. What is modified?

Line 68: "To test QED..." is really jargon for " to "test QED-based theoretical calculations"

Lin 76: It would be nice to give the theoretical value for χ here.

Line 116: More should be said about the "literature value" for the magnetic moment of ^{209}Bi . I

assume the issue is with the theory (diamagnetic corrections) that relates the measurements to the moment of the bare nucleus – that should be made explicit here.

Line 118: include the half-life of ^{208}Bi .

Line 144: I do not understand how the “hyperfine puzzleopens the route to high precision measurements in ion traps”.

Methods:

An obvious criticism of the hyperfine splitting measurements is that, because of the very large Doppler shifts, they require an accurate determination of the ion beam velocity. The situation is made worse by the fact that they use a co-propagating beam for the Li-like and counter-propagating beam for the H-like, so systematic error in the beam velocity add constructively in the HFS difference. The ion beam velocity is determined from the velocity of the electrons in the electron cooler, allowing for such effects as space charge and surface potentials, and velocity shifts due to buncher phasing. My impression is that within the rather large group of authors there is enough expertise in these issues that the uncertainty estimates are likely to be reasonable. But the possibility of an unaccounted for, or under-estimated systematic error remains. A measurement using co- and counter propagating beams for both ion species, see for example DeVore et al, Physical Review Letters 100, 243001 (2008) (a measurement on highly-charged silicon ions), would be more reliable.

Line 184: “ Gaussian lineshape as expected for a Doppler broadened line..”. The line shape may be well fitted by a Gaussian, but only if the ion beam velocity has a Gaussian distribution, not just because it is Doppler broadened.

Reviewer #3 (Remarks to the Author):

The authors report on high-precision measurements of the hyperfine structure of hydrogenlike and lithium like Bismuth ions at the GSI storage ring. This allows a sensitive test of QED in the strong magnetic field regime, according to a proposal by Shabaev and coworkers: namely, a well-chosen linear combination of the H-like and Li-like splitting eliminates theoretical uncertainties from the imprecise knowledge of the magnetization distribution in the nucleus. The experimental result is found to differ from the theoretical prediction by about 7 times the combined experimental and theoretical uncertainties.

To achieve such a high precision (improved by more than order of magnitude with respect to first results by the same authors published in 2014 and 2015) with such highly charged ions is in itself an experimental feat. The observed discrepancy with strong-field bound-state QED calculations is very interesting and will undoubtedly trigger further theoretical and experimental studies. Only time will tell if this “hyperfine puzzle” takes on as much importance as the one concerning the proton radius, but these results, which test a different sector of QED, are certainly worthy of being communicated to a broad readership.

In general the paper is well written and the results are clearly presented. My only qualm is about the justification of the experimental uncertainty budget (Table 2) which I feel is a bit too brief. Especially in view of the large deviation from theory, it is important to explain how the various systematic shifts have been tested and estimated. While a discussion about some of these shifts can be found in earlier papers by the authors (especially Ref. 31), it would be useful to include a brief summary of the methods used to estimate them (possibly as online supplementary material, if space limitations prevent from including it into the main text).

One question: could additional measurements on one (or more) other charge state(s) of ^{209}Bi be of

some use, with the idea of cross-checking the magnetic moment of the nucleus? If yes, this could be mentioned in the last paragraph of the paper.

Reviewers' comments:

Reviewer #1 (Remarks to the Author):

This is a very important paper which demonstrates the biggest deviation from QED so far, at the level of 7 sigma (previous deviations in the muon g-factor, proton and deuterium radii are at the smaller scale but have raised an enormous interest in the scientific community). Current work is based on the brilliant idea that the nuclear uncertainty cancels out in the combination of the single-electron and 3 electron hyperfine structures.

Whatever reason for the deviation is - new physics, or unknown problems of the QED theory, or even some unknown experimental uncertainty - the paper must be published and very carefully studied by the whole physics community. This work gives a very strong motivation for further theoretical and experimental studies.

V. Flambaum.

Nothing to comment. We are very glad that Prof. Flambaum regards the manuscript as very important.

Reviewer #2 (Remarks to the Author):

The manuscript presents measurements of the ground state hyperfine splittings in hydrogen-like and lithium-like ^{209}Bi using a fast ion beam laser resonance technique. A certain combination of the splittings, which is theoretically predicted to be insensitive to the nuclear structure of the ^{209}Bi nucleus (which otherwise shifts the splittings through the Bohr-Weisskopf effect), is found to differ from the theoretical prediction for this combination by about 7 times the combined experimental and theoretical uncertainties. The authors conclude that the discrepancy could indicate a problem with the theory, most likely that the cancellation of the Bohr-Weisskopf effect does not occur as expected, or else that the literature values of the nuclear magnetic moment of ^{209}Bi may be in error, (presumably because of a problem with the theory that enables nuclear magnetic moments to be obtained from measurements on neutral ^{209}Bi atoms). In the abstract, they also indicate that the discrepancy may indicate a more fundamental problem with the theory of Quantum Electrodynamics. I think this particular claim is unduly speculative, albeit making the abstract more enticing to the non-expert reader. But overall, I find the manuscript scientifically sound (but see my detailed comments),

interesting, well written, and suitable for publication in Nature Communications.

Detailed comments:

Abstract:

Although most of the abstract suggests that “fundamental theory of QED” is being tested” (which in any case is a rather old fashioned choice of words, since QED, as part of electroweak theory, is no longer “fundamental”),-the real issue is the theory of electron-nuclear interactions – the nucleus, or even the proton, being an enormously more complicated object than the electron. So I suggest that the last sentence in the abstract be given greater emphasis. (Line 40 should also have “charge radius of the “deuteron”, not of “deuterium”).

We have changed the last sentence of the abstract in order to follow the advice:

“Our result joins a series of recent measurements that challenge the theory of electron-nuclear interactions like the muon g-factor [6], the proton radius [7] and, most recently, the charge radius of the deuteron [8].“

Lines 57-59: Sentence structure: There should be commas after “regime” and “does not account for”.

But also “ does not account for, or is modified..” is unclear. What is modified?

The point has been made more clear by changing the sentence to

“New effects might appear in the interaction of the electron with itself, the vacuum or the nuclear fields in this regime, i.e., the hyperfine interaction might be affected by the existence of new particles not included yet in the current standard model and therefore not considered in state-of-the-art QED calculations.”

Line 68: “To test QED...” is really jargon for “ to “test QED-based theoretical calculations”

We have changed this accordingly.

Lin 76: It would be nice to give the theoretical value for Xi here.

Done.

Line 116: More should be said about the “literature value” for the magnetic moment of ^{209}Bi . I assume the issue is with the theory (diamagnetic corrections) that relates the measurements to the moment of the bare nucleus – that should be made explicit here.

We have put this explicitly:

“Possible reasons could be that the magnetic moment of the ^{209}Bi nucleus is considerably different from the accepted value in the literature due to incorrectly determined diamagnetic-shielding or chemical-shift corrections for the NMR data or that the cancellation of the BW-effect does not work as expected.”

Line 118: include the half-life of ^{208}Bi .

Done.

Line 144: I do not understand how the “hyperfine puzzleopens the route to high precision measurements in ion traps”.

Methods:

An obvious criticism of the hyperfine splitting measurements is that, because of the very large Doppler shifts, they require an accurate determination of the ion beam velocity. The situation is made worse by the fact that they use a co-propagating beam for the Li-like and counter-propagating beam for the H-like, so systematic error in the beam velocity add constructively in the HFS difference. The ion beam velocity is determined from the velocity of the electrons in the electron cooler, allowing for such effects as space charge and surface potentials, and velocity shifts due to buncher phasing. My impression is that within the rather large group of authors there is enough expertise in these issues that the uncertainty estimates are likely to be reasonable. But the possibility of an unaccounted for, or under-estimated systematic error remains. A measurement using co-and counter propagating beams for both ion species, see for example DeVore et al, Physical Review Letters 100, 243001 (2008) (a measurement on highly-charged silicon ions), would be more reliable.

Response:

Indeed, a determination of the transition frequencies in both directions would lead to a better accuracy as demonstrated in the reference provided by the referee as well as similar previous work of the same group. We have been aware of this possibility and have used it previously ourselves for on-line measurements of short-lived Be isotopes [Nörtershäuser et al. PRL 102, 062503 (2009)]. But unfortunately it is not feasible for heavy H-like or Li-like Bi due to the inconvenient Doppler shifts in the other directions. It would require strong laser pulses at a wavelength of 100 nm for copropagating excitation of H-Like Bi and about 3.5 micrometer for counterpropagating excitation of Li-like Bi. Measurements at much lower velocities would shift the excitation wavelength into more accessible regions of the spectrum but impede the fluorescence detection, especially for Li-like Bi due to the inefficient detection of infrared photons.

Line 184: “ Gaussian lineshape as expected for a Doppler broadened line..”. The line shape may be well fitted by a Gaussian, but only if the ion beam velocity has a Gaussian distribution, not just because it is Doppler broadened.

The referee is right and we have changed the text accordingly to

“The signal shape is well reproduced with a Gaussian lineshape as indicated by the structureless fitting residuals shown in the lowest frame of each sub-figure. This was expected for an ion beam described by a Maxwell-Boltzmann velocity distribution established by intra-beam scattering and interactions with the electron beam.”

Reviewer #3 (Remarks to the Author):

The authors report on high-precision measurements of the hyperfine structure of hydrogen like and lithium like Bismuth ions at the GSI storage ring. This allows a sensitive test of QED in the strong magnetic field regime, according to a proposal by Shabaev and coworkers: namely, a well-chosen linear combination of the H-like and Li-like splitting eliminates theoretical uncertainties from the imprecise knowledge of the magnetization distribution in the nucleus. The experimental result is found to differ from the theoretical prediction by about 7 times the combined experimental and theoretical uncertainties.

To achieve such a high precision (improved by more than order of magnitude with respect to first results by the same authors published in 2014 and 2015) with such highly charged ions is in itself an experimental feat. The observed discrepancy with strong-field bound-state QED calculations is very interesting and will undoubtedly trigger further theoretical and experimental studies. Only time will tell if this “hyperfine puzzle” takes on as much importance as the one concerning the proton radius, but these results, which test a different sector of QED, are certainly worthy of being communicated

to a broad readership.

In general the paper is well written and the results are clearly presented. My only qualm is about the justification of the experimental uncertainty budget (Table 2) which I feel is a bit too brief.

Especially in view of the large deviation from theory, it is important to explain how the various systematic shifts have been tested and estimated. While a discussion about some of these shifts can be found in earlier papers by the authors (especially Ref. 31), it would be useful to include a brief summary of the methods used to estimate them (possibly as online supplementary material, if space limitations prevent from including it into the main text).

We have added a few sentences for each systematic that has been studied to the material in the methods section.

One question: could additional measurements on one (or more) other charge state(s) of ^{209}Bi be of some use, with the idea of cross-checking the magnetic moment of the nucleus? If yes, this could be mentioned in the last paragraph of the paper.

In principle it would be possible to use other charge states but the splitting decreases quickly with $1/n^3$ and will not be easily measurable directly with lasers anymore. Instead a measurement of the nuclear and the electronic g-factor in H-like Bi using a combination of laser optical pumping and RF spectroscopy as it is being prepared at ARTEMIS (see manuscript) will shed new light on this issue.

REVIEWERS' COMMENTS:

Reviewer #2 (Remarks to the Author):

All the points raised in my first review have been addressed satisfactorily.

A minor point is that the last sentence of the abstract has a repeated "and". It would read better as "...and have found a 7-sigma discrepancy with (or "compared to", but not just "to") the theoretical prediction."

Reviewer #3 (Remarks to the Author):

The authors have followed my main recommendation and included a discussion of systematic uncertainties.

Concerning the Abstract, I note that the actual manuscript file differs from what is described in the rebuttal:

"We have changed the last sentence of the abstract in order to follow the advice:

"Our result joins a series of recent measurements that challenge the theory of electron-nuclear interactions like the muon g-factor [6], the proton radius [7] and, most recently, the charge radius of the deuteron [8]."

In fact, this sentence has also been moved from the Abstract to the end of the Introduction, which I think is a good decision, since (following the remark by Reviewer #2) this claim could be slightly misleading without a presentation of all possible explanations for the discrepancy. The message could perhaps be even clearer if it is moved to the Section 3 (e.g. just after the comment of lines 113-116).

Apart from this optional suggestion, I recommend publication of this manuscript in its present form.

The remaining referee comments were taken into account:

Reviewer #2 (Remarks to the Author):

All the points raised in my first review have been addressed satisfactorily.

A minor point is that the last sentence of the abstract has a repeated "and". It would read better as "...and have found a 7-sigma discrepancy with (or "compared to", but not just "to") the theoretical prediction."

Done

Reviewer #3 (Remarks to the Author):

The authors have followed my main recommendation and included a discussion of systematic uncertainties.

[...]

"Our result joins a series of recent measurements that challenge the theory of electron-nuclear interactions like the muon g-factor [6], the proton radius [7] and, most recently, the charge radius of the deuteron [8]."

In fact, this sentence has also been moved from the Abstract to the end of the Introduction, which I think is a good decision, since (following the remark by Reviewer #2) this claim could be slightly misleading without a presentation of all possible explanations for the discrepancy. The message could perhaps be even clearer if it is moved to the Section 3 (e.g. just after the comment of lines 113-116). Apart from this optional suggestion, I recommend publication of this manuscript in its present form.

We prefer to keep the sentence at its current position.